# SMC condensin entraps chromosomal DNA by an ATP hydrolysis dependent loading mechanism in *Bacillus subtilis*

**Larissa Wilhelm[1], Frank Bürmann[1], Anita Minnen[1], Ho-Chul Shin[2†], Christopher P Toseland[1], Byung-Ha Oh[2], Stephan Gruber[1]***

[1]Chromosome Organization and Dynamics, Max Planck Institute of Biochemistry, Martinsried, Germany; [2]Department of Biological Sciences, KAIST Institute for the Biocentury, Cancer Metastasis Control Center, Korea Advanced Institute of Science and Technology, Daejeon, Republic of Korea

**Abstract** Smc–ScpAB forms elongated, annular structures that promote chromosome segregation, presumably by compacting and resolving sister DNA molecules. The mechanistic basis for its action, however, is only poorly understood. Here, we have established a physical assay to determine whether the binding of condensin to native chromosomes in *Bacillus subtilis* involves entrapment of DNA by the Smc–ScpAB ring. To do so, we have chemically cross-linked the three ring interfaces in Smc–ScpAB and thereafter isolated intact chromosomes under protein denaturing conditions. Exclusively species of Smc–ScpA, which were previously cross-linked into covalent rings, remained associated with chromosomal DNA. DNA entrapment is abolished by mutations that interfere with the Smc ATPase cycle and strongly reduced when the recruitment factor ParB is deleted, implying that most Smc–ScpAB is loaded onto the chromosome at *parS* sites near the replication origin. We furthermore report a physical interaction between native Smc–ScpAB and chromosomal DNA fragments.

*For correspondence: sgruber@ biochem.mpg.de

Present address: †Functional Genomics Research Center, Korea Research Institute of Bioscience and Biotechnology, Daejeon, Republic of Korea

**Competing interests:** The authors declare that no competing interests exist.

## Introduction

Compaction and individualization of sister DNA molecules is a prerequisite for efficient segregation of the genetic material to daughter cells during cell division. Multi-subunit Structural Maintenance of Chromosomes (SMC) protein complexes—such as cohesin and condensin—are major determinants of chromosome structure and dynamics during the cell cycle in eukaryotes as well as in prokaryotes (*Hirano, 2006*; *Thadani et al., 2012*; *Gruber, 2014*). Condensin subunits were initially identified as abundant, non-histone components of mitotic chromosomes in metazoans (*Hirano and Mitchison, 1994*). In mitosis, condensin localizes together with topoisomerase II in punctate structures to the longitudinal core of chromatids, called the chromosome axis (*Coelho et al., 2003*; *Maeshima and Laemmli, 2003*; *Ono et al., 2004*). Inactivation of condensin subunits by mutation or depletion results in severe morphological aberrations and mechanical sensitivity of metaphase chromosomes, and subsequently to defects in their segregation during anaphase (*Hirano and Mitchison, 1994*; *Ono et al., 2003*; *Gerlich et al., 2006*). In bacteria, Smc–ScpAB is the prevalent version of SMC protein complexes. Its distant relatives MksBEF and MukBEF can be found scarcely scattered over most of the bacterial phylogenetic tree and in isolated branches of proteobacteria, respectively (*Gruber, 2011*). In *Bacillus subtilis* and *Streptococcus pneumoniae*, Smc–ScpAB is recruited to a region around the replication origin by ParB/Spo0J protein bound to *parS* sites, thereby forming a discrete focus—also called condensation center—on each nascent copy of the chromosome (*Gruber and Errington, 2009*; *Sullivan et al., 2009*; *Minnen et al., 2011*). Inactivation of Smc–ScpAB in *B. subtilis* under nutrient rich

**eLife digest** The genome of any living organism holds all the genetic information that the organism needs to live and grow. This information is written in the sequence of the organism's DNA, and is often divided into sub-structures called chromosomes. Different species have different sized genomes, but even bacteria with some of the smallest genomes still contain DNA molecules that are thousand times longer than the length of their cells. DNA molecules must thus be highly compacted in order to fit inside the cells. DNA compaction is particularly important during cell division, when the DNA is being equally distributed to the newly formed cells.

In plants, animals and all other eukaryotes, large protein complexes known as condensin and cohesin play a major role in compacting, and then separating, the cell's chromosomes. Many bacteria also have condensin-like complexes. At the core of all these complexes are pairs of so-called SMC proteins. However, it is not clear how these SMC proteins direct chromosomes to become highly compacted when cells are dividing.

Wilhelm et al. have now developed two new approaches to investigate how SMC proteins associate with bacterial DNA. These approaches were then used to study how SMC proteins coordinate the compaction of chromosomes in a bacterium called *Bacillus subtilis*. The experiments revealed that SMC proteins are in direct physical contact with the bacterial chromosome, and that bacterial DNA fibers are physically captured within a ring structure formed by the SMC proteins.

Wilhelm et al. suggest that these new findings, and recent technological advances, have now set the stage for future studies to gain mechanistic insight into these protein complexes that organize and segregate chromosomes.

growth conditions blocks separation of sister replication origins and consequentially leads to lethal defects in chromosome partitioning (*Gruber et al., 2014*; *Wang et al., 2014*). Smc–ScpAB thus promotes the initial stages of chromosome segregation in *B. subtilis*, likely by condensing and individualizing the emerging copies of the chromosome in preparation for their segregation to opposite halves of the cell.

The canonical SMC complex in bacteria comprises five subunits: (1) two Smc proteins, which each form a 45 nm long antiparallel coiled coil that connects an ABC-type ATPase 'head' domain at one end of the coiled coil with a 'hinge' homodimerization domain at the other end (*Hirano et al., 2001*), (2) a single ScpA subunit, which belongs to the kleisin family of proteins and associates via its C-terminal winged-helix domain (WHD) with the bottom 'cap' surface of one Smc head and via its N-terminal helical domain with the 'neck' coiled coil region of the other Smc protein (*Bürmann et al., 2013*), and (3) a dimer of ScpB protein, which binds to the central region of ScpA (*Bürmann et al., 2013*; *Kamada et al., 2013*). Overall, the pentameric Smc–ScpAB complex displays a highly extended conformation harboring a central channel, which is surrounded by a closed tripartite ring formed by the Smc dimer and the ScpAB$_2$ sub-complex. The *B. subtilis* Smc coiled coils associate with one another to form rod-shaped Smc dimers (*Soh et al., 2015*). Furthermore, the Smc head domains can interact directly with one another—via a composite interface that includes two molecules of ATP. Binding to ATP, head engagement and ATP hydrolysis likely control and drive the biochemical action of Smc–ScpAB.

Models for SMC condensation activity have been proposed based on observations made with isolated SMC dimers, SMC fragments or holo-complexes. Such protein preparations support the bridging of given DNA molecules in vitro as indicated by the re-annealing of single stranded DNA, intermolecular DNA ligation, DNA catenation and the co-purification of labeled and unlabeled DNA molecules (*Sutani and Yanagida, 1997*; *Losada and Hirano, 2001*; *Cui et al., 2008*). Many SMC complexes bound to different segments of DNA might thus come together and anchor DNA in condensation centers or at the chromosome axis. Oligomeric assemblies of bacterial Smc proteins have indeed been observed by Atomic Force Microscopy and Electron Microscopy (*Mascarenhas et al., 2005*; *Fuentes-Perez et al., 2012*). This model provides a straightforward explanation for the compaction activity of SMC. However, it is unclear how such apparently indiscriminate DNA aggregation would promote rather than block the individualization of sister chromosomes (*Gruber, 2014*). Local wrapping of DNA around the SMC complex could result in well-defined lengthwise

condensation of DNA. However, too little SMC protein appears to be present in chromosomes to yield decent levels of compaction by simple wrapping. A different hypothesis is based on the finding that the structurally related cohesin complex holds sister chromatids in eukaryotes together by entrapping sister DNA fibers within its ring (*Gruber et al., 2003*; *Gligoris et al., 2014*). Accordingly, individual SMC complexes might entrap and expand loops of DNA, thereby driving lengthwise condensation of chromosomes with little limitations in the attainable levels of compaction (*Nasmyth, 2001*; *Alipour and Marko, 2012*).

Here, we investigate how the prokaryotic SMC–kleisin complex binds to chromosomes in vivo using a novel whole-chromosome assay.

## Results

### A chromosome entrapment assay

We initially attempted to detect topological interactions between *B. subtilis* Smc–ScpAB and plasmid DNA using pull-down assays as previously described (*Ivanov and Nasmyth, 2005*; *Ghosh et al., 2009*; *Cuylen et al., 2011*). However, several attempts failed to provide clear evidence for entrapment of small circular DNA by prokaryotic condensin. Conceivably, Smc–ScpAB does not interact with these artificial substrates in a physiological manner. To circumvent this possibility, we established an inverse assay by immobilizing whole chromosomes of *B. subtilis* in agarose plugs and monitoring their association with covalently closed rings of Smc–ScpA under harsh protein denaturing conditions (*Figure 1A*). To develop the chromosome entrapment assay we first performed experiments with the replicative sliding clamp, DnaN, in *B. subtilis*, which is known to entrap DNA in a topological manner. Furthermore, most of cellular DnaN protein is maintained in the vicinity of active replication forks in *B. subtilis*, presumably by its topological association with leading and lagging strand DNA (*Su'etsugu and Errington, 2011*).

Based on the crystal structure of *S. pneumoniae* DnaN we engineered a pair of cysteine residues (N114C, V313C) into the *B. subtilis* protein so that DnaN can be cross-linked into covalent rings in the presence of a cysteine-specific cross-linker such as BMOE (*Figure 1B*). For detection a cys-less variant of the HaloTag ('HT') was fused to the C-terminus of DnaN (*Figure 1—figure supplement 1B*) and the construct was integrated into the genome of *B. subtilis* via allelic replacement at the endogenous locus. The *dnaN-ht* genes with and without cysteine mutations supported normal growth of *B. subtilis*, implying that they encoded functional DnaN proteins (data not shown). In vivo cross-linking of DnaN-HT resulted in two additional, slow migrating bands in SDS-PAGE gels (*Figure 1C*), corresponding to single and double cross-linked species of DnaN dimers, designated as X-DnaN-HT and XX-DnaN-HT, respectively (*Figure 1—figure supplement 1A*). We next embedded cells in agarose plugs and disrupted their cell walls by lysozyme digestion. Agarose plugs were then subjected to an electric field in the presence of SDS to denature and remove any unattached proteins from chromosomes. Plugs were finally treated with benzonase to digest genomic DNA and to release any stably entrapped protein. DnaN-HT protein was then analysed by in-gel fluorescence. Non-crosslinkable DnaN-HT was efficiently depleted from agarose plugs during the entrapment assay (*Figure 1C*). In contrast, the double cross-linked, circular form of DnaN(N114C, V313C)-HT (XX-DnaN-HT) was retained in the agarose plug during electrophoresis with high efficiency (∼50% of input). A minor fraction of single cross-linked DnaN dimer (X-DnaN-HT) was also observed. This is likely generated from XX-DnaN-HT by spontaneous hydrolysis of thiol-malemide adducts during protein isolation (*Kalia and Raines, 2007*; *Baldwin and Kiick, 2013*). Importantly, the presence of benzonase during cell lysis eliminated all DnaN from the plug, indicating that circular DnaN is retained in plugs via its interaction with cellular DNA. Furthermore, in the absence of the cross-linker BMOE, no DnaN-HT was detected in the eluate fraction (*Figure 1—figure supplement 1C*). The chromosome entrapment assay thus specifically detects a topological association of intact chromosomes with DNA sliding clamps and confirms that a major fraction (at least 50%) of DnaN is loaded onto DNA in rapidly growing cells.

### Prokaryotic condensin entraps chromosomal DNA

Next, we used the newly developed chromosome entrapment assay to test for an association between native chromosomes and Smc–ScpAB complexes. Cysteine pairs were introduced at the Smc–Smc and at both Smc–ScpA interfaces and a HT was fused at the C-terminus of Smc to allow in-gel fluorescence detection (*Figure 2A*) (*Bürmann et al., 2013*). Strains bearing the cysteine mutations and the

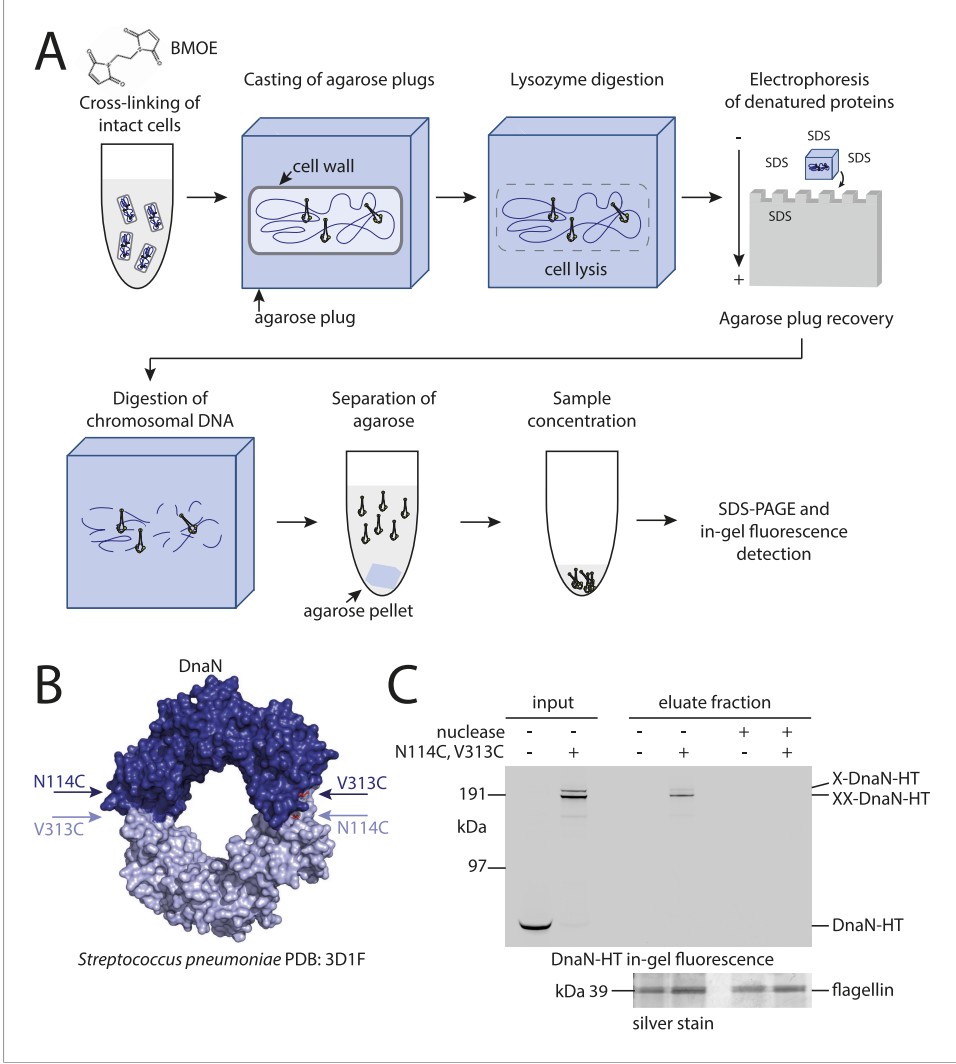

**Figure 1**. Development of the chromosome entrapment assay using DnaN. (**A**) Scheme for the chromosome entrapment assay. Cells are incubated with the cysteine cross-linker BMOE, lysed in agarose plugs and subjected to an electric field in the presence of SDS buffer. Proteins stably bound to chromosomal DNA are re-isolated from nuclease treated agarose plugs, concentrated and analyzed by SDS-PAGE. (**B**) Crystal structure of *S. pneumoniae* DnaN (PDB: 3D1F) in surface representation. The monomers of DnaN are shown in dark and light blue colours, respectively. The positions of an engineered pair of cysteine residues (N114C and V313C) at the monomer–monomer interface of *B. subtilis* DnaN are indicated by arrows. (**C**) Chromosome entrapment by DnaN. Cells of strains BSG1449 (*dnaN-HT*) and BSG1459 (*dnaN(N114C, V313C)-HT*) were cross-linked with BMOE and subjected to the chromosome entrapment assay. Input and eluate fractions were analysed by in-gel detection of fluorescently labeled HT fused to DnaN (top panel). Eluate fractions of samples treated with or without nuclease during cell lysis are indicated as nuclease '+' or '−', respectively. Eluate fractions were further analyzed by silver staining revealing that another protein was consistently co-isolated during the chromosome entrapment assay (bottom panel). This protein—identified as flagellin by mass spectrometry—was retained independently of the integrity of the chromosome. The following figure supplement is available: *Figure 1—figure supplement 1*: DNA entrapment by DnaN.

The following figure supplement is available for figure 1:

**Figure supplement 1**. DNA entrapment by DnaN.

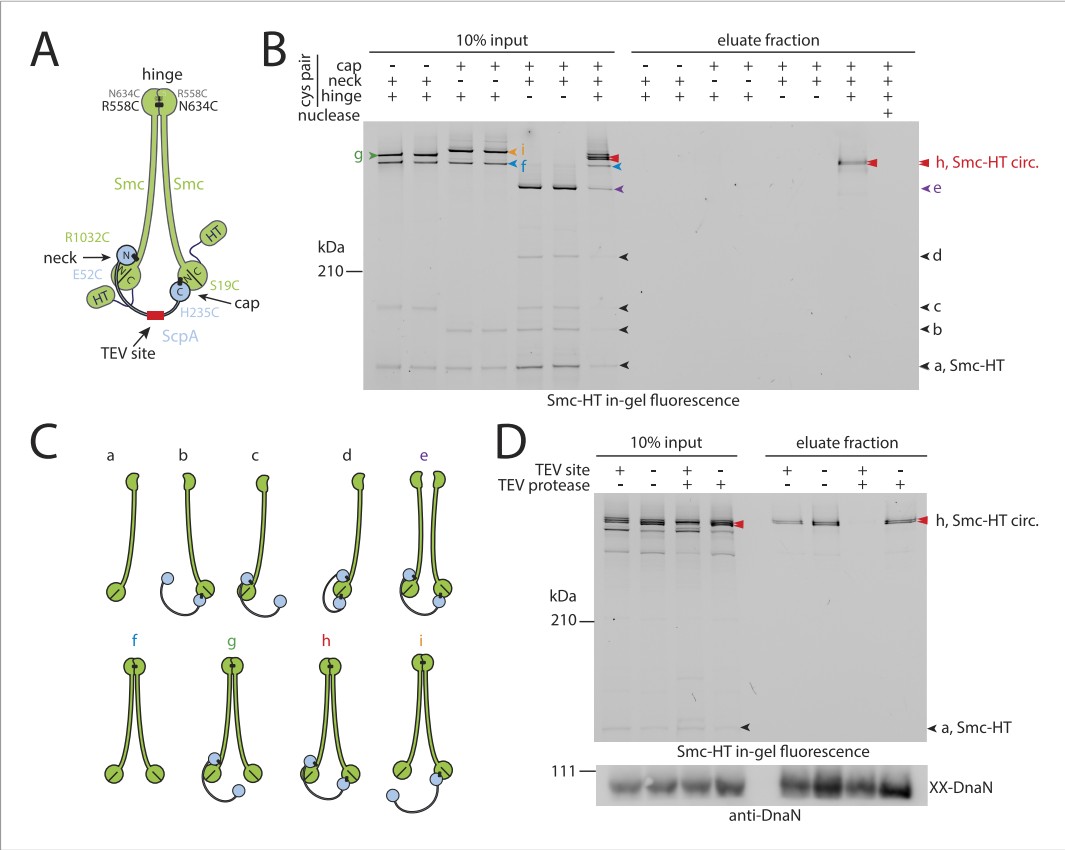

**Figure 2**. Prokaryotic condensin entraps the chromosome. (**A**) Scheme for the cross-linking of Smc-HaloTag ('HT') and ScpA into a covalent Smc–ScpA–Smc ring. (**B**) Chromosome entrapment of covalent Smc₂–ScpA rings. Cells of strains BSG1782, BSG1809-1813 and BSG1831 were cross-linked and subjected to the chromosome entrapment assay. Cross-linked Smc-HT species were visualized by in-gel fluorescence detection. The presence or absence of cysteine pairs at each of the three ring interfaces are indicated by '+' and '−', respectively. An aliquot of cells of strains BSG1782 was incubated with benzonase during cell lysis (nuclease '+'). The positions of uncross-linked Smc-HT and fully cross-linked, circular Smc–ScpA–Smc species are indicated by 'Smc-HT' and 'Smc-HT circ.'; all species are labelled by colour-coded arrowheads (see panel **C** for legend). Circular species ('h') are labeled by a double pointed arrowhead. (**C**) Schematic depiction of the structure of cross-linked Smc–ScpA species ('a'–'i'). (**D**) TEV cleavage of ScpA prevents entrapment of Smc–ScpAB in agarose plugs. In-gel fluorescence detection of Smc-HT derived from strains BSG1807 and BSG1832. The presence or absence of TEV sites in ScpA and of TEV protease during cell lysis is indicated by '+' and '−', respectively. Cleavage of ScpA(TEVs) by TEV protease creates new species of cross-linked Smc-HT (see 'input' samples) and prevents entrapment of Smc-HT in agarose plugs (see 'eluate fraction') (top panel). 'XX-DnaN' serves as internal assay control visualized by immunoblotting of cross-linked species of DnaN protein (bottom panel). The following figure supplement is available: *Figure 2—figure supplement 1*: DNA entrapment by wild-type Smc–ScpAB (I) and *Figure 2—figure supplement 2*: DNA entrapment by wild-type Smc–ScpAB (II).

The following figure supplements are available for figure 2:

**Figure supplement 1**. DNA entrapment by wild-type Smc–ScpAB (I).

**Figure supplement 2**. DNA entrapment by wild-type Smc–ScpAB (II).

Smc-HaloTag fusion supported normal growth on nutrient rich medium demonstrating the functionality of the modified Smc complex (*Figure 2—figure supplement 1A*). Cells were treated with BMOE and extracts were analysed by SDS-PAGE. As internal control for the chromosome entrapment assay we employed the DnaN(N114C, V313C) protein, whose double cross-linked form was detected in input and eluate samples by immunoblotting (*Figure 2—figure supplement 1B*). Various species of Smc–ScpAB were identified in extracts of BMOE cross-linked cells by in-gel

fluorescence. These correspond to fully cross-linked Smc–ScpA–Smc rings and several intermediate cross-linking species as reported previously (*Figure 2B,C*) (*Bürmann et al., 2013*). To reveal the identity of all species, strains lacking one of six engineered cysteines were used as controls that collectively form several intermediate cross-linked species but no fully cross-linked rings of Smc–ScpAB (*Figure 2—figure supplement 1C*). In these control samples little or no Smc-HT protein was retained in agarose plugs under denaturing conditions as expected for any non-circular protein (*Figure 2B*). In the presence of all pairs of cysteine, however, a set of two closely migrating species was consistently detected at significant levels after the chromosome entrapment assay (~10–20% of input material) (*Figure 2B*). We argued that the two closely migrating species might correspond to Smc–ScpAB with a single or a double cross-link at the Smc hinge. Consistent with this notion we find that only a single species of $Smc_2$–ScpA accumulated during the chromosome entrapment assay when a single cysteine residue (R643C) was used to cross-link the Smc hinge domains (*Figure 2—figure supplement 2A*). These findings strongly suggest that Smc–ScpAB is bound to chromosomes via entrapment of chromosomal DNA. If this were indeed the case, then its retention in agarose plugs should depend on the integrity of Smc–ScpAB rings and of chromosomal DNA. Incubation of agarose plugs with the nuclease benzonase during cell lysis eliminated the Smc-HT and DnaN signal in the sample (*Figure 2B*, *Figure 2—figure supplement 1B*). To disrupt covalent Smc–ScpAB rings, we inserted cleavage sites for TEV protease into the linker region preceding the C-terminal WHD of ScpA and incubated cells during lysozyme treatment with recombinant TEV protease to open any circular $Smc_2$–ScpA species. As expected, little or no Smc-HT signal was detected in agarose plugs after TEV cleavage of ScpA (*Figure 2D*). To exclude any artefacts due to the presence of the HT on Smc we have repeated the chromosome entrapment assay with an untagged allele of Smc using immunoblotting with anti-Smc antibodies for the detection of cross-linked species, which yielded very similar results (*Figure 2—figure supplement 2C*). Furthermore, we found that $Smc_2$-ScpA rings are stably trapped in agarose plugs over extended periods of time in constant or alternating electric fields (data not shown). Thus, our chromosome entrapment assay specifically detects the association between intact chromosomal DNA and rings of Smc–ScpAB in *B. subtilis*, demonstrating that DNA fibers pass through the Smc ring.

## A full Smc ATPase cycle is required for loading of condensin onto chromosomes

Next, we established the requirements for the formation of interconnections between Smc–ScpAB rings and chromosomes. The intrinsic ATPase activity of cohesin has previously been implicated in stable association with chromosomes (*Arumugam et al., 2003*; *Weitzer et al., 2003*). More specifically, ATP hydrolysis has been hypothesized to transiently open an entry gate for DNA in the cohesin ring during its loading onto chromosomes (*Gruber et al., 2006*; *Hu et al., 2011*). To test what steps of the ATP hydrolysis cycle in Smc–ScpAB are involved in the entrapment of chromosomal DNA, we made use of *smc* alleles harboring mutations that specifically prevent ATP binding (K37I), engagement of Smc head domains (S1090R) or ATP hydrolysis (E1118Q) (*Figure 3A*) (*Hirano and Hirano, 2004*). The three mutant proteins are expressed at normal levels in *B. subtilis* being indicative of proper protein folding (*Figure 3—figure supplement 1A*). However, they do not support growth on nutrient rich medium similar to *smc* null mutants, implying that all steps of the ATPase cycle are essential for Smc functionality (*Figure 3—figure supplement 1B*) (*Gruber et al., 2014*). For the chromosome entrapment assay, these Smc ATPase mutations were combined with cysteine mutations for BMOE cross-linking. To support their viability, the resulting strains as well as the wild-type controls were grown in minimal medium. The three mutant Smc proteins assembled into normal Smc–ScpAB complexes as judged by Smc–ScpA cross-linking, albeit there is a slight decrease in the fraction of ScpA proteins bridging Smc dimers and a concomitant minor increase in ScpA subunits bound to single Smc proteins (*Figure 3—figure supplement 1C*, species 'e' and 'd', respectively) (*Bürmann et al., 2013*). Intriguingly, the ATP binding and engagement mutants abolished the fraction of covalent ring species retained in the agarose plug during the chromosome entrapment assay (*Figure 3B*). In case of the ATP hydrolysis mutant Smc(E1118Q) only minute amounts of cross-linked rings were recovered from SDS treated plugs. This small fraction of stably bound condensin conceivably arises as a consequence of residual levels of ATP hydrolysis activity in Smc(E1118Q) (*Hirano and Hirano, 2004*). Thus, ATP binding and ATP dependent Smc head engagement—and most probably also ATP hydrolysis—are essential for entrapment of chromosomal DNA by condensin

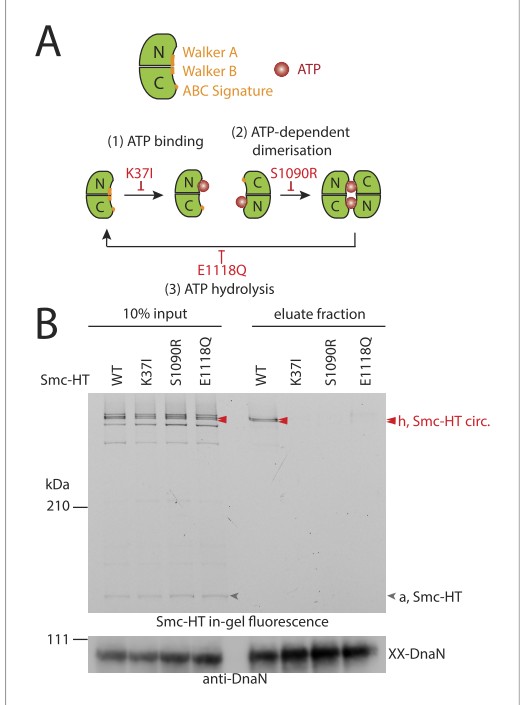

**Figure 3**. The Smc ATPase is required for loading of DNA into Smc–ScpAB. (**A**) A scheme for the ATP hydrolysis cycle of Smc. Schematic positions for Walker A, Walker B and ABC-signature motifs on the Smc head domain are shown (top row). ATP binding to the Walker A domain is blocked in Smc(K37I) '(1)'. ATP-dependent engagement of two Smc heads is abolished in the Smc (S1090R) mutant '(2)'. The E1118Q mutation strongly reduces ATP hydrolysis '(3)'. (**B**) Smc ATPase mutations abolish chromosomal loading of Smc–ScpAB. In-gel fluorescence detection of Smc-HT of input and eluate fractions from a representative chromosome entrapment assay performed with strains BSG1782 and BSG1784-6. Protein extracts (10% of input) were loaded next to samples subjected to the entrapment assay. Selected cross-linked species of Smc-HT are labeled (top panel). Detection of cross-linked species of DnaN by immunoblotting was used as internal assay control (bottom panel). The following figure supplement is available: *Figure 3—figure supplement 1*: ATPase mutants of Smc–ScpAB.
The following figure supplement is available for figure 3:

**Figure supplement 1**. ATPase mutants of Smc-ScpAB.

in bacteria, as has been supposed for cohesin in yeast. Furthermore, the strict requirement of several steps of the ATPase cycle strongly suggests that entrapment of DNA corresponds to the physiological form of association with the bacterial chromosome.

## ScpB and ParB proteins are essential for normal loading of condensin onto chromosomes

What other factors might be required for the loading of condensin onto DNA? The ScpB subunit forms homodimers that bind in an asymmetric manner to the central region of a single ScpA monomer. It thus is in close proximity of the Smc ATPase domains. Together with ScpA it putatively plays a role in the regulation of the Smc ATPase activity (*Kamada et al., 2013*). Its precise molecular function, however, is not clear yet. To test whether ScpB is involved in the association of Smc–ScpA rings with chromosomes we combined the cysteine mutations in Smc and ScpA with an *scpB* in-frame deletion (*Figure 4—figure supplement 1*). Ring formation was only mildly affected by the absence of ScpB as judged by BMOE cross-linking and in-gel fluorescence detection (*Figure 4A*) (*Bürmann et al., 2013*). However, Smc complexes lacking ScpB subunits failed to entrap chromosomes altogether demonstrating that ScpB is absolutely required for loading of prokaryotic condensin onto chromosomal DNA.

ParB proteins—bound to *parS* sites—are crucial for efficient targeting of Smc–ScpAB to a large region of the chromosome near the replication origin (*Gruber and Errington, 2009*; *Sullivan et al., 2009*; *Minnen et al., 2011*). ParB might act by simply increasing the local concentration of Smc–ScpAB around *oriC* either before or after its loading onto the chromosome. Alternatively, ParB bound to *parS* sites might be more directly involved in the loading reaction itself, for example, as catalytic factor, and its absence might thus affect levels of chromosomal condensin. To test this, we performed the chromosome entrapment assay with cells lacking the *parB* gene. Intriguingly, the levels of Smc–ScpAB entrapping chromosomal DNA were strongly reduced in the *parB* null mutant as judged by the limited retention of Smc–ScpA species in agarose plugs (*Figure 4B*). Thus, ParB protein likely promotes the entrapment of chromosomal DNA by Smc–ScpAB. This strongly suggests that most condensin is loaded onto the chromosome at *parS* sites, where ParB protein is bound. In all other parts of the chromosome entrapment of DNA fibers by Smc–ScpAB might be very inefficient. The cysteine bearing *smc* allele causes growth defects when combined with *ΔparB* (*Figure 4—figure supplement 1*). Therefore, we cannot formally exclude the possibility that the decreased loading of Smc observed in *ΔparB* are due to the cysteine

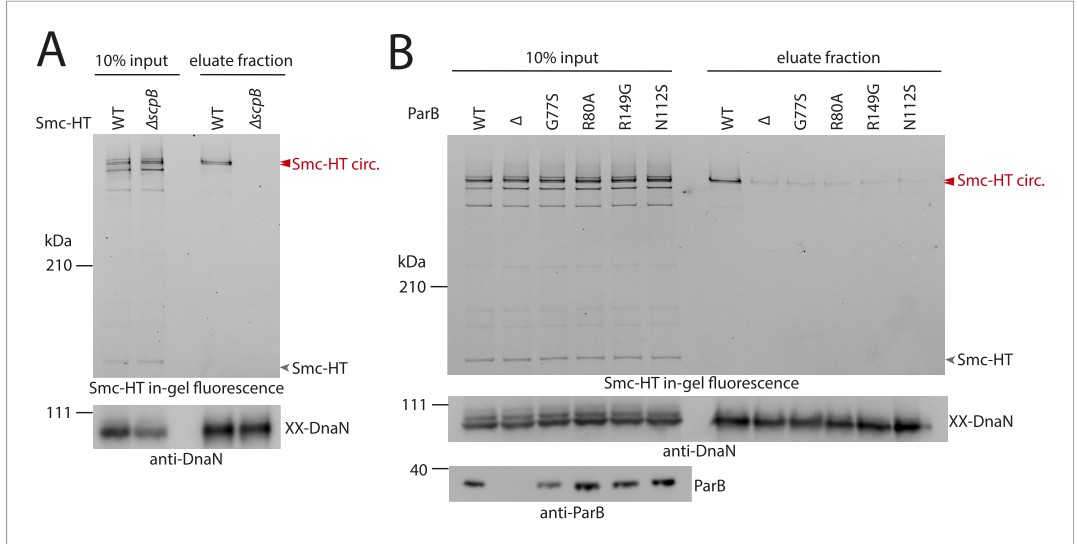

**Figure 4**. ScpB and ParB are essential for efficient DNA entrapment by Smc complexes. (**A**) Deletion of *scpB* eliminates loading of chromosomal DNA into Smc complexes. In-gel fluorescence detection of Smc-HT in input and eluate fractions is shown from chromosome entrapment assays performed with strains BSG1782 ('WT') and BSG1850 ('*ΔscpB'*) (top panel). DnaN was used as internal control (bottom panel). (**B**) Several *parB* mutations interfere with efficient chromosomal loading of Smc–ScpAB. Input and eluate fractions from chromosome entrapment assays with strains BSG1782, BSG1783 and BSG1960-3 were analysed by in-gel fluorescence detection of Smc-HT (top panel). DnaN was used as internal control (middle panel). Immunoblotting using polyclonal rabbit anti-ParB antiserum confirms near-normal expression of mutant ParB proteins (bottom panel). The following figure supplement is available: *Figure 4—figure supplement 1*: Growth of *smc*, *parB* double mutants.

The following figure supplement is available for figure 4:

**Figure supplement 1**. Growth of *smc(Cys)* mutants.

---

modifications in Smc and that chromosomal loading of wild-type Smc is not or much less affected by *parB* deletion.

Previously, two *parB* point mutations (N112S and R149G), which prevent the formation of Smc-GFP foci, have been isolated in *B. subtilis* (**Gruber and Errington, 2009**). We found that these mutations strongly impair loading of Smc onto the chromosome in the entrapment assay similar to *ΔparB* (**Figure 4B**). The R149G mutation is positioned on the helix-turn-helix motif of ParB and might thus directly affect binding to *parS* sites (**Leonard et al., 2004**). The N112S mutation, however, is located in another highly conserved region, which has been implicated in the 'spreading' of ParB protein from *parS* sequences into adjacent DNA (**Leonard et al., 2004**; **Graham et al., 2014**). The spreading of ParB along several kb of DNA is a feature conserved in plasmid and chromosome derived ParB proteins, however, the underlying mechanism is only poorly understood (**Rodionov et al., 1999**). It might possibly involve the formation of a large nucleoprotein complex (**Broedersz et al., 2014**). Several other mutants of ParB (including *B. subtilis* ParB G77S and R80A) have been reported to be defective in spreading from *parS* sites (**Breier and Grossman, 2007**; **Graham et al., 2014**). Intriguingly, also these mutations resulted in largely reduced levels of Smc on the chromosome in our entrapment assay, being comparable to the levels found in a *parB* deletion mutant (**Figure 4B**). This implies that ParB spreading from *parS* sites or formation of large nucleoprotein complexes might be essential for loading of DNA into the Smc ring by ParB. These findings are consistent with the observation that formation of Smc-GFP foci near the origin of replication are affected by the G77S mutation (**Sullivan et al., 2009**). In summary, these results demonstrate that several factors—including ScpB protein, a ParB/*parS* nucleoprotein complex and the Smc ATPase cycle—are required to promote efficient loading of condensin rings onto the chromosome.

## Smc–ScpAB rings physically associate with chromosomal DNA fragments

Smc proteins and fragments thereof exhibit affinity for single- and double-stranded DNA in vitro (*Chiu et al., 2004*; *Hirano and Hirano, 2006*; *Soh et al., 2015*). The physical contacts with DNA might occur once condensin has been successfully loaded onto chromosomes and thus be a permanent feature of chromosomal Smc–ScpAB. Alternatively, the direct association with DNA might be restricted to certain intermediates in the chromosomal loading reaction. To test for interactions between Smc–ScpAB and specific chromosomal DNA fragments, we have affinity-purified endogenous Smc–ScpAB from *B. subtilis* cell lysates using a short Avitag peptide fused to the C-terminus of the Smc protein, which gets biotinylated when the biotin ligase gene *birA* is co-expressed ('Smc-Avitag'). We then examined fractions for the co-purification of fragments of chromosomal DNA—generated by restriction digest with XbaI—using quantitative PCR with primer pairs specific for different parts of the chromosome. Since we worried that Smc–ScpAB might not be sufficiently stable in diluted cell extracts, we cross-linked the three ring interfaces in Smc–ScpAB using BMOE cross-linking of engineered pairs of cysteines. A small fraction of chromosomal DNA was reproducibly co-purified with wild-type Smc-Avitag, whereas the yield of co-purified DNA was significantly improved by the presence of cross-linkable cysteine residues in Smc–ScpAB (*Figure 5*). In both cases origin-proximal regions (*yyaD*, *parS-359*, *dnaA* and *dnaN*) of the chromosome were more efficiently enriched than distal regions (*amyE*, *trnS* and *ter*) by the co-purification with Smc implying that the observed Smc-DNA contacts are dependent on chromosomal loading of Smc–ScpAB by ParB protein at *parS* sites and are thus physiologically relevant (*Figure 5*, *Figure 5—figure supplement 1*) (*Gruber and Errington, 2009*). The association of DNA with wild-type and BMOE cross-linked Smc–ScpAB was highly sensitive to washes with a salt solution (2M NaCl), suggesting that it was dependent on electrostatic contacts between DNA and protein. These DNA contacts are presumably formed by the Smc–ScpAB complex itself. Alternatively, albeit less likely, other chromosomal proteins physically bound to DNA could prevent the release of condensin from DNA by blocking its sliding towards DNA ends.

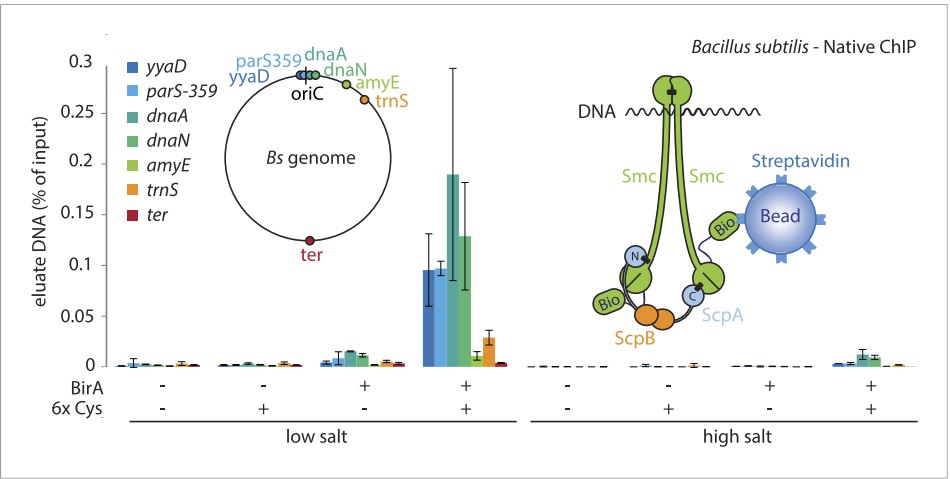

**Figure 5**. A physical interaction of Smc–ScpAB rings and chromosomal DNA. Co-purification of chromosomal DNA fragments with native Smc–ScpAB. Cells of strains BSG1104-5 and BSG1107-8 were treated with the cross-linker BMOE prior to cell lysis. Strains carrying ('+') or lacking ('−') cysteine mutations ('6xCys') in the Smc-AviTag construct were expressed in presence ('+') or absence ('−') of the biotin ligase ('BirA'). Beads were washed in the presence of either a 150 mM ammonium acetate buffer ('low salt') or a 2 M sodium chloride buffer ('high salt'). The co-purification of DNA fragments with Smc-biotin on streptavidin beads was measured by quantitative PCR using primer pairs specific for genomic positions indicated on a representation of the circular *B. subtilis* genome. Mean values and standard deviations were calculated from two independent biological replicates. The following figure supplement is available: *Figure 5—figure supplement 1*: Chromatin immuno-precipitation of Smc.

The following figure supplement is available for figure 5:

**Figure supplement 1**. Chromatin immuno-precipitation of Smc.

## Discussion

### The agarose entrapment assay

In many cases, it is challenging to measure the activity and outcome of biochemical processes in the living cell. Here, we report the establishment of a straight-forward method to determine the physical association of ring-shaped protein complexes with whole bacterial chromosomes. Two examples, the SMC condensin complex and the sliding clamp DnaN, document the significant potential of our simple entrapment approach. In principle, similar assays should also be possible with eukaryotic cells and for many other chromosomal proteins such as for example hexameric helicases and certain transcription factors. Furthermore, analogous procedures might be useful to address biological questions related to other denaturation-resistant cellular structures such as cell wall polymers (e.g., made up of peptidoglycan, chitin or cellulose).

### DNA entrapment by an ancestral SMC–kleisin complex

SMC–kleisin complexes are major governors of chromosome superstructure in most branches of the phylogenetic tree. The eukaryotic variants cohesin and condensin have been suggested to work as concatenases, which hold selected stretches of DNA together by simple embracement in their ring (*Haering et al., 2008*; *Cuylen et al., 2011*). Whether DNA entrapment is an ancestral and thus fundamentally conserved function of SMC–kleisin complexes, however, remained elusive so far. Furthermore, interaction studies developed for cohesin and condensin are based on small, artificial DNA substrates and might thus not necessarily reflect the mode of binding to native chromosomes. These assays also fall short of providing an estimate for the fraction of SMC complexes involved in interlocked associations with DNA and thus leave open the possibility that DNA entrapment might be an insignificant side reaction. Finally, it has not been tested under physiological conditions, whether the ATPase cycle is required for proper loading of DNA into any SMC–kleisin complex. To provide answers to these questions, we have established the chromosome entrapment assay to determine the association of prokaryotic condensin with native chromosomes. Our results clearly demonstrate that chromosomal DNA is loaded into condensin complexes in *B. subtilis*—in a manner that depends on the non-SMC subunit ScpB and at least one full cycle of Smc ATPase activity. The chromosome entrapment assay recovers about 10–20% of the fully cross-linked input material. This number probably understates the real proportion of chromosomally entrapped Smc complexes due to the loss of material during protein re-isolation from agarose plugs, due to possible adverse effects of cysteine mutations on Smc–ScpAB loading and because cysteine-maleimide linkages are vulnerable to hydrolytic reversal during and after the entrapment assay (*Kalia and Raines, 2007*; *Baldwin and Kiick, 2013*). Interestingly, a recent single-molecule tracking study in *B. subtilis* revealed two major populations of Smc: 80% of Smc-YFP proteins are displaying highly dynamic behavior on the nucleoid, whereas the other 20% (and most ScpA-YFP protein) are immobile and constrained within a small volume of the cell (*Kleine Borgmann et al., 2013*). This immobile fraction possibly represents Smc–ScpAB complexes embracing origin proximal DNA after loading at *parS* sites as observed in the entrapment assay.

Our results show that embracement of chromosomal DNA is a predominant feature of Smc–ScpAB, which has been evolutionarily retained in cohesin and likely all other SMC–kleisin complexes as well. Since the chromosome entrapment assay is based on the immobilization of intact replicating chromosomes, which possibly represent internally knotted and branched DNA structures, it is conceivable that Smc–ScpAB rings are linked to chromosomal DNA by non-topological capture of DNA loops, which themselves might be interlinked (i.e., knotted) with other parts of the chromosome. Therefore, it remains to be determined whether DNA entrapment by Smc–ScpAB is of topological (*Figure 6A*) and/or non-topological (*Figure 6B*) nature.

### How might entrapment of DNA at ParB/*parS* nucleoprotein complexes promote sister DNA segregation?

Smc–ScpAB plays a crucial role in the segregation of replication origins in *B. subtilis* cells (*Gruber, 2014*; *Wang et al., 2014*), presumably by organizing nascent sister chromosomes so that their spatial overlap and entanglement is minimized. It is tempting to speculate that ParB/*parS* not only enriches Smc in the vicinity of the replication origin but also sets up lengthwise compaction of chromosomes by

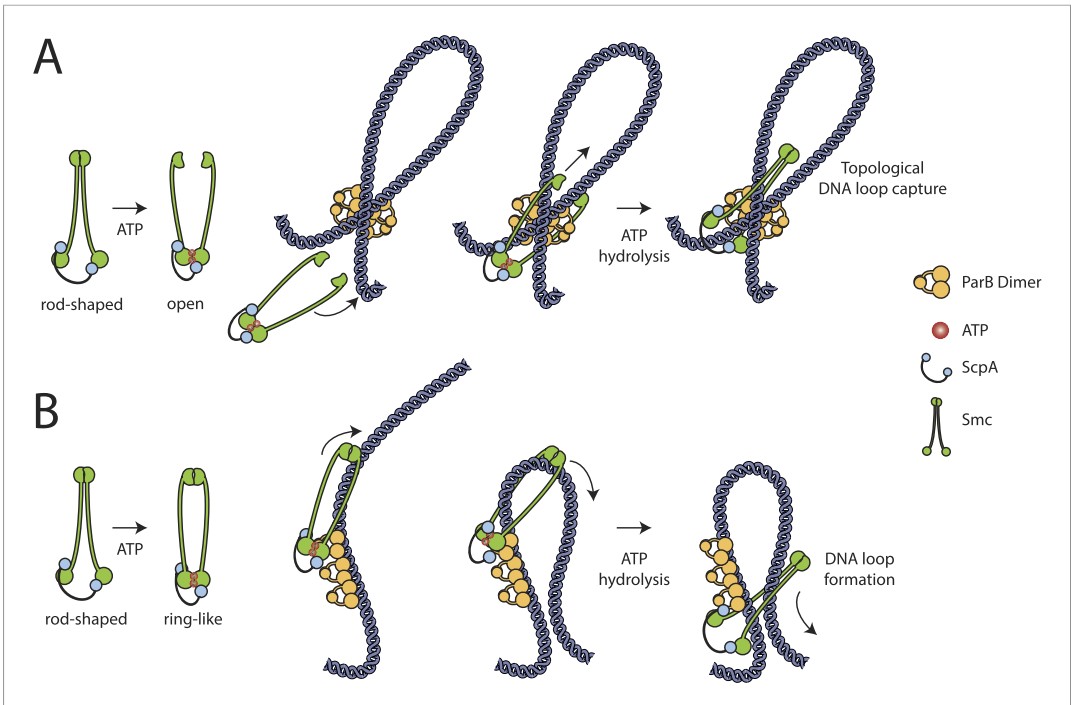

**Figure 6**. Models for entrapment of chromosomal DNA by Smc–ScpAB. (**A**) Loop capture model. DNA loops might be pre-formed within ParB/*parS* nucleoprotein assemblies. Driven by ATP dependent engagement of Smc head domains Smc–ScpAB adopts a ring-like configuration. Occasional opening of the Smc hinge then allows capture of ParB-DNA loops within Smc–ScpAB. Subsequent ATP hydrolysis by Smc locks the hinge in a closed state and stabilizes the structure. (**B**) Loop formation model. ParB/*parS* might serve as a landing platform for Smc–ScpAB allowing Smc–ScpAB in its ring-like conformation to guide DNA into its central cavity. Continuous extrusion of DNA through Smc–ScpAB then drives lengthwise condensation of chromosomes. Ring opening is not required in this model and DNA entrapment by Smc–ScpAB is thus strictly non-topological here. The following figure supplement is available: *Figure 6—figure supplement 1*: Quantitative Blotting of Smc protein and *parB* DNA.

The following figure supplement is available for figure 6:

**Figure supplement 1**. Quantitative Blotting of Smc protein and *parB* DNA.

presenting DNA of a certain topology to the Smc–ScpAB complex. Consistent with this notion, we found that several *parB* mutants, which are defective in the ability to form large nucleoprotein complexes and to spread from *parS* sites, fail to promote loading of Smc–ScpAB onto the chromosome. Thus, Smc–ScpAB might capture, stabilize and expand structures–such as DNA loops or coils–that are pre-formed within larger ParB/*parS* nucleoprotein assemblies (*Figure 6A*). Alternatively, ParB/*parS* might serve as an elaborate landing platform on the chromosome, where Smc–ScpAB initiates with the help of ParB the lengthwise compaction of chromosomes by forming and extruding loops of DNA (*Figure 6B*). Extrusion of DNA might involve the translocation of Smc–ScpAB along DNA fibers made up of naked or ParB coated DNA. These models are not mutually exclusive. A reasonable first step towards understanding the architecture of Smc/ParB/*parS* assemblies might be the investigation of physical interactions between ParB/*parS* and Smc–ScpAB and their functional interconnection with the Smc ATPase cycle. In the future, Smc/ParB/*parS* structures could serve as a relatively simple paradigm for chromosome organization by the more intricately regulated cohesin and condensin complexes in eukaryotes.

## The role of ScpB and ATP hydrolysis in Smc–ScpAB loading

ParB serves a supplemental—albeit important—role in the Smc loading process. In contrast, Smc functionality and its loading onto DNA in vivo is critically dependent on the ScpB subunit and the

ATPase cycle. It remains unclear though what the exact role of ScpB and ATP hydrolysis in the entrapment of DNA might be. If loading indeed depends on transient opening of a DNA entry gate, the open state would likely represent an energetically unfavorable reaction intermediate (*Figure 6A*). Timely opening would require energy input as well as tight regulation. We have recently demonstrated that the ATP dependent engagement of Smc head domains—together with DNA binding to the Smc hinge domain—can transform the configuration of the Smc coiled coil from a rod to a more open ring-like conformation (*Soh et al., 2015*). Hydrolysis of ATP and/or ScpB binding could drive a subsequent conformational change that might open the SMC–kleisin ring (*Figure 6A*). Alternatively, ScpB and/or ATP hydrolysis might stabilize Smc–ScpAB once loaded onto DNA or promote Smc's sliding along DNA to allow efficient extrusion of DNA loops by the Smc–ScpAB ring (*Figure 6B*).

## How do so few Smcs organize so much DNA?

Deletion or mutation of the *parB* gene results in a clear drop in the levels of chromosomally bound condensin (∼5–10 fold less) in our chromosome entrapment assay (*Figure 4B*, *Figure 2—figure supplement 2C*). In addition to this loading defect, also the specific recruitment of Smc–ScpAB towards the replication origin is lost in the absence of ParB (*Gruber and Errington, 2009*; *Sullivan et al., 2009*). Thus, in *parB* mutants only a very small proportion of cellular Smc–ScpAB is bound to chromosomes within the replication origin region, where it presumably performs its essential function by promoting the separation of nascent sister chromosomes (*Gruber et al., 2014*; *Wang et al., 2014*). Nevertheless, defects in chromosome segregation are rather mild in *parB* mutants when compared to mutants of *smc*. Using quantitative blotting of Smc protein and replication origin DNA from cell extracts, we have estimated the average number of Smc protein to be around 30 dimers per replication origin in a fast growing population of cells (*Figure 6—figure supplement 1*). Assuming that all Smc complexes entrap chromosomal DNA in wild-type cells, only three to six Smc dimers (10–20% of total) are loaded onto the chromosome in *parB* mutants according to our measurements. Thus, a handful of Smc–ScpAB complexes, which are presumably randomly distributed over the chromosome, appears to be capable of supporting near-normal chromosome segregation under these conditions—when chromosome segregation is already compromised by the loss of the *parABS* system. Few Smc–ScpAB therefore seem to be able to provide enough organization to the replication origin region and the remainder of the chromosome to prevent lethal accumulation of inter-linked sister chromosomes. It is conceivable that individual Smc–ScpAB complexes are able to organize large chunks of a bacterial chromosome, possible by forming giant loops of DNA. Alternatively, Smc activity might be needed only at a limited number of defined locations on the *B. subtilis* chromosome and/or for very short periods of time. However, when levels of functional Smc dimers are in addition reduced for example by hypomorphic mutations in the *smc* gene itself, the loss of ParB protein becomes lethal (*Gruber and Errington, 2009*).

This work reveals the mode of association of Smc–ScpAB with bacterial chromosomes, highlights its striking evolutionary conservation and demonstrates the involvement of the SMC ATPase cycle in chromosomal loading. Future work must address the underlying biochemical mechanisms to get basic insight into the architectural role of SMC in chromosome biology.

## Materials and methods

### *B. subtilis* strains and media

Genetic modifications at *smc*, *scpAB*, *parB* and *dnaN* loci were generated via double cross-over recombination in strains derived from *B. subtilis* 1A700 or *B. subtilis* 168ED. Genotypes of strains used in this study are listed in *Supplementary file 1*. Cells were transformed with plasmids or *B. subtilis* genomic DNA using a 2-step starvation protocol as previously described (*Hamoen et al., 2002*; *Bürmann et al., 2013*). Transformants were selected by growth on nutrient agar (NA) plates (Oxoid, UK) supplemented with antibiotics as required: 5 µg ml$^{-1}$ kanamycin, 80 µg ml$^{-1}$ spectinomycin, 10 µg ml$^{-1}$ tetracycline, 5 µg ml$^{-1}$ chloramphenicol, 1 µg ml$^{-1}$ erythromycin and 25 µg ml$^{-1}$ lincomycin. Strains displaying a condensin null phenotype were selected on SMG medium instead: SMM salt solution (2 g l$^{-1}$ ammonium sulphate, 14 g l$^{-1}$ dipotassium hydrogen phosphate, 6 g l$^{-1}$ potassium dihydrogen phosphate, 1 g l$^{-1}$ trisodium citrate, 0.2 g l$^{-1}$ magnesium sulphate, 6 g l$^{-1}$ potassium hydrogen phosphate) supplemented with 5 g l$^{-1}$ glucose, 20 mg l$^{-1}$ tryptophan and 1 g l$^{-1}$ glutamate

with the respective antibiotics. Strains were single-colony purified and grown in the absence of antibiotics for experiments.

## Colony formation assay

Cells were pre-grown in a 96-well plate in SMG medium for 24 hr at 37°C. Overnight cultures were diluted 9^2-fold (high density spots) or 9^5-fold (low density spots) and spotted onto NA or SMG agar plates. Plates were incubated at 37°C for 12 hr on NA or 24 hr on SMG agar.

## Growth conditions and in vivo cysteine cross-linking

Cells were grown in either LB Miller medium (10 g $l^{-1}$ tryptone, 5 g $l^{-1}$ yeast extract, 10 g $l^{-1}$ sodium chloride) or SMG medium to mid-exponential phase at 37°C (in LB Miller medium, $OD_{600}$ of 0.4; in SMG medium, $OD_{600}$ of 0.03). Cells were harvested by centrifugation or vacuum filtration and washed in ice-cold PBS supplemented with 0.1% glycerol ('PBSG'). Cell aliquots (corresponding to 1 ml at an $OD_{600}$ of 1.25) were re-suspended in PBSG and incubated with the cross-linker BMOE (bis-maleimidoethane, Applichem, Germany) at a concentration of 1 mM (diluted from a 20 mM stock solution in DMSO). After a 10 min incubation on ice the reaction was quenched by addition of 2-mercaptoethanol ('2-ME') to a final concentration of 28 mM.

For the preparation of protein extracts ('input') a mixture of following components was added to an aliquot of cells: 400 units ready-lyse lysozyme (Epicentre, Madison, WI), 12.5 units benzonase (Sigma-Aldrich, St. Louis, MO) and a protease-inhibitor cocktail (Sigma). In addition, 1 µM HT Oregon Green substrate (Promega, Madison, WI) was added to cell suspensions with HaloTag bearing alleles. Samples were then incubated for 20 min at 37°C protected from light. Finally, the samples were heated to 70°C for 5 min in LDS Sample Buffer (NuPage, Thermo Scientific, Waltham, MA) containing 200 mM DTT and loaded onto a SDS-PAGE gel (see below). Gels with Oregon Green labeled samples were scanned on a Typhoon scanner (GE Healthcare, UK) with Cy3-DIGE filter setup.

## Chromosome entrapment assay

Cells were grown, cross-linked and quenched as described above. Lysozyme stock solution, protease inhibitor and HT substrate were added to an aliquot of cells at concentrations given above. The cell suspension was mixed immediately in a 1:1 ratio with a 2% solution of Megabase agarose (BioRad, Hercules, CA) or low-melt agarose (BioRad) and casted into 100 µl agarose plugs using plug molds (BioRad). Agarose plugs were incubated for 20 min at 37°C, protected from light, and then loaded into the wells of a 6% SDS-PAGE Tris-glycine gel. The polyacrylamide mini-gel was run for 60 min at 25 mA protected from light. Agarose plugs were then re-extracted from the gel and transferred into 1.5 ml Eppendorf tubes. 1 ml of Wash Buffer ('WB': 0.01 mM EDTA, 0.5 mM Tris, 0.5 mM $MgCl_2$, 0.01% SDS) was added per agarose plug. Plugs were incubated for 10 min with gentle agitation protected from light. This step was repeated once. Wash buffer was then discarded and replaced by 100 µl fresh WB supplemented with 50 units of benzonase (Sigma). Plugs were incubated at 37°C for 30 min. Plugs were melted at 85°C for 2 min under vigorous agitation. The samples were frozen at −80°C and stored overnight. Samples were then thawed, centrifuged for 10 min at 4°C and 14,000×$g$ and transferred to a 0.45 µm Cellulose acetate spin column (Costar, Tewksbury, MA) and spun for 1 min at 10,000×$g$. The flow-through was concentrated in a Speed Vac (Thermo Scientific, no heating, 2.5 hr running time). The concentrated sample was resuspended in LDS Sample Buffer (NuPage) containing 200 mM DTT and heated for 3 min at 70°C. Samples were loaded onto Tris-acetate gels (3–8% Novex, Thermo Scientific) and run for 2.5 hr at 35 mA per gel at 4°C. For DnaN detection Bis-Tris gels (8–12% Novex) were run for 1 hr at 200 V at room temperature. Gels were either scanned on a Typhoon scanner (FLA 9000, GE Healthcare) with Cy3-DIGE filter setup or immuno-blotted using antibodies against DnaN or Smc (see below). For cleavage of ScpA(TEVs) or degradation of chromosomal DNA 15 units of His-TEV protease or 12.5 units of benzonase, respectively, was added before casting agarose plugs.

## Co-purification of chromosomal DNA fragments with Smc-AviTag protein

*B. subtilis* strains containing *smc-tev-avitag* alleles were grown to $OD_{600}$ of 0.4 in 100 ml LB Miller at 37°C. Part of the culture was fixed with formaldehyde and subjected to chromatin immunoprecipitation (ChIP) as described by (*Gruber and Errington, 2009*) using a rabbit anti-Smc antiserum. In parallel, 10 ml of the culture were mixed with ice, harvested by centrifugation and washed in cold

PBSG. Cells were resuspended in 200 µl PBSG and treated with 0.5 mM BMOE for 10 min on ice. The reaction was quenched with 14 mM 2-ME and cells were washed once in CutSmart buffer (New England Biolabs, Ipswich, MA). Cells were resuspended in 200 µl CutSmart containing 10 kU Ready-Lyse lysozyme (Epicentre), 40 U XbaI (New England Biolabs) and a protease inhibitor cocktail (Sigma). The suspension was incubated for 15 min at 37°C before addition of 1800 µl buffer LS (10 mM Tris/HCl, 150 mM NH$_4$OAc, 1 mM EDTA, 6 mM 2-ME, 0.05% Tween-20, 0.01% NaN$_3$, final pH 7.9 at 23°C). Lysates were centrifuged for 5 min at 20,000×$g$. Subsequently, 1400 µl of the extract were incubated with 100 µl Dynabeads Streptavidin C1 for 30 min at room temperature. Beads were washed once in buffer LS, then split, resuspended either in buffer LS or in buffer HS (10 mM Tris/HCl, 2 M NaCl, 1 mM EDTA, 6 mM 2-ME, 0.05% Tween-20, 0.01% NaN$_3$, final pH 7.9 at 23°C) and incubated for 15 min at room temperature. Beads were washed twice with buffer LS, and protein/DNA complexes were eluted for 1 hr at 22°C by incubation with 350 µl LS containing TEV protease and 1 mM DTT. DNA from input and eluate fractions was purified by treatment with 0.5 mg/ml Proteinase K for 1 hr at 55°C followed by phenol/chloroform extraction. Samples were analysed by quantitative PCR using the second derivative maximum of a four parameter logistic model similar to the method described by (*Zhao and Fernald, 2005*).

## Immunoblotting and antibodies

After gels were scanned for in-gel fluorescence detection, they were immediately transferred onto a PVDF membrane (Immobilon-P, Merck Millipore, Germany) using semi-dry transfer. Membranes were blocked with 3.5% (wt/vol) milk powder in PBS with 0.1% Tween 20. Rabbit polyclonal sera against *B. subtilis* DnaN (*Lenhart et al., 2013*), *B. subtilis* Smc (this paper) and *B. subtilis* ParB (this paper) were used as primary antibodies for immunoblotting at dilutions of 1:5000 each. The membrane was developed with HRP-coupled secondary antibodies and chemiluminescence (Super-Signal West Femto, Thermo Scientific) and visualized on a LAS-3000 scanner (FujiFilm, Germany).

## DnaN cross-linking time-course

To estimate DnaN cross-linking kinetics (*Figure 1—figure supplement 1A*) samples were grown as described above. An aliquot of cells was incubated with the cross-linker BMOE (1 mM) for the indicated length of time before the reaction was quenched with 2-ME (28 mM).

## Estimation of cellular Smc Protein and *parS-359* DNA

### Protein purification and quantification

The expression plasmid for unmodified wild-type Smc was a gift from Mark Dillingham (Uni. of Bristol, UK). Wild-type Smc protein was expressed and purified as described in (*Fuentes-Perez et al., 2012*) with an additional Superose 6 10/300 GL (GE Healthcare) gel filtration added as a final step in the purification. Gel filtration was performed in storage buffer 50 mM Tris–HCl at pH 7.5, 150 mM NaCl, and 1 mM DTT. The concentration of purified untagged Smc protein was determined by measuring the absorption of the protein at 280 nm in 6 M guanidine chloride (*Grimsley and Pace, 2004*). An extinction coefficient (51,230 M$^{-1}$ cm$^{-1}$ at 280 nm) for the *B. subtilis* Smc protein was obtained using the ProtParam tool at www.expasy.org.

### Spike-in PCR product and Southern probe

The spike-in DNA was generated by PCR using wild-type genomic DNA preparation as template DNA and forward ('STG246': cttgcgatttttgcttctcc; complementary to the *yyaD* locus) and reverse primers ('STH602': ttatcgtgcgaaagcagttg; complementary to the *gidA* locus) producing a DNA fragment of 7238 bp in size covering the *parS-359* site within the *parB* gene. The PCR product was purified using a PCR-purification kit (QIAquick PCR purification kit, Qiagen, Germany) and its concentration was measured by absorption at 260 nm on a Nanodrop 2000c (Thermo Scientific) photometer. The molecular weight was calculated based on the base composition of the DNA. For generation of the *parS-359*-specific Southern probe a PCR with primers annealing within and downstream of the *parB* locus ('STG301': acatgagaattcgttttttcatttatgattctcgttcagacaaaagctc and 'STK534': gcaatctgcagcatggcattcttcag) was performed on a wild-type genomic DNA preparation generating a 714 bp long PCR product. This PCR DNA was used as a template for a second PCR for random incorporation of digoxigenin ('DIG') labelled nucleotides following the 'random PCR DIG labelling protocol' (Roche, Germany).

## Cell culture and harvesting

Wild-type *B. subtilis* strain BSG1001 and the doubly modified strain BSG2058 were grown in LB Miller medium (10 g l$^{-1}$ tryptone, 5 g l$^{-1}$ yeast extract, 10 g l$^{-1}$ sodium chloride) to mid exponential phase at 37°C (OD$_{600}$ of 0.3). Cells were harvested by centrifugation and washed in ice-cold PBS supplemented with 0.1% glycerol ('PBSG').

## Protein extracts for quantitative Western blotting

Protein extracts were prepared from 1 ml of a cell suspension at OD$_{600}$ of 2. Cells of BSG1001 and BSG2058 were pelleted and resuspended in 50 µl PBSG and mixed in the appropriate ratios. Cells were lysed by addition of a mix of following enzymes: 400 units ready-lyse lysozyme (Epicentre), 12.5 units benzonase (Sigma) and a protease-inhibitor cocktail (Sigma) in a total volume of 5 µl. After 30 min incubation at 37°C the purified Smc protein was spiked into the whole cell lysates in a total volume of 10 µl as given in *Figure 6—figure supplement 1B*. Finally, the samples were heated to 70°C for 5 min in LDS Sample Buffer (NuPage) containing 100 mM DTT. 1/20 of the final protein extracts were loaded onto a SDS-PAGE gel. For immunoblotting and antibodies see 'Materials and methods'.

## Genomic DNA preparation for quantitative Southern blotting

Bacterial cell cultures were identical to the ones used for Quantitative Western blotting. Aliquots of BSG1001 and BSG2058 were taken (equivalent to 1 ml of culture at OD$_{600}$ of 4) and resuspended in 95 µl 50 mM EDTA pH 8.0. 5 µl lysozyme was added to a final concentration of 0.5 mg/ml and samples were incubated for 30 min at 37°C. 500 µl of a commercially available Lysis Buffer ('Nuclei Lysis Solution', Wizard Genomic DNA Purification Kit, Promega) was added, followed by 5 min incubation at 80°C. Samples were incubated with a final concentration of 0.05 mg/ml RNAse A for 10 min at 37°C. Then each sample was sonicated very gently (4× 0.1 s pulses at lowest power setting, Bandelin 'Sonoplus', Germany) to solubilize chromosome fragments. Cell lysates of BSG1001 and BSG2058 were mixed and purified PCR product was spiked in as given in the figure. 200 µl of 'Protein Precipitation Solution' (Wizard Genomic DNA Purification Kit, Promega) was added to each sample followed by a 10 s vortexing step and 5 min incubation on ice. Samples were spun 3 min at 13,000×*g* and supernatant was transferred into a fresh 1.5 ml tube containing 600 µl 100% isopropanol. The tubes were inverted 20 s until the DNA precipitated and DNA was pelleted by centrifugation for 2 min at 13,000×*g*. Pellets were washed with 70% ethanol and finally resuspended in 41 µl 10 mM Tris-Cl pH 8.5. Restriction digest was performed using 80 units PstI enzyme (NEB) per reaction for 1 hr at 37°C and the enzyme was inactivated for 20 min at 80°C. 1/3 of each preparation (equivalent to 1 ml OD$_{600}$ of 1.3) was loaded onto a 0.6% Megabase agarose (BioRad) gel containing 1 µg/ml Ethidiumbromide (Sigma) in 44.5 mM Tris, 44.5 mM Boric acid, 1 mM EDTA ('0.5× TBE Buffer'). Gel was run 16 hr at 1V/cm. Transfer for Southern Blotting was performed using an alkaline buffer (1.5 M NaCl, 0.4 N NaOH) for 24 hr onto a nylon membrane ('Hybond-N', GE Healthcare Life Sciences). Hybridisation was done using the digoxigenin-labelled Southern probe (see above) specific to the *parS359* locus in a commercial hydridisation buffer ('DIG Easy Hyb Granules', Roche) for 4 hr at 42°C. Stringency washes, blocking and detection was performed following the 'CDP-Star' Manual (Roche, Cat.No. 12 041 677 001).

## Data quantification and calculation

The intensity of the individual wt Smc and wt *parS-359* bands from the Western and Southern blots, respectively, were quantified using ImageJ 1.48v software and values were plotted against the calculated concentrations of Smc protein and *parS-359* PCR DNA in each sample. The concentration of Smc protein and *parS-359* DNA in wild-type extracts was determined from the intensity of the Smc/*parS-359* band using a linear fit of the standard curve.

## Acknowledgements

We are very grateful to Lyle Simmons for providing *B. subtilis dnaN-gfp* strains and α-DnaN antiserum and to Mark Dillingham for sharing expression plasmids and purification protocols for untagged Smc protein. We thank Boris Pfander and Stefan Jentsch for sharing resources, Anna Reisenbichler for technical help and all members of the Gruber laboratory for stimulating discussions and helpful comments on the manuscript. This work was supported by an ERC Starting Grant (DiseNtAngle, #260853) to SG and by the Max Planck Society and by the National Research Foundation of Korea (NRF) grant funded by the Korea government (#2014-022694) to B-HO.

## Additional information

### Funding

| Funder | Grant reference | Author |
|---|---|---|
| European Research Council (ERC) | ERC StG 260853 | Stephan Gruber, Larissa Wilhelm, Frank Bürmann, Anita Minnen, Christopher P Toseland |
| Max-Planck-Gesellschaft | | Stephan Gruber |
| National Research Foundation of Korea | 2013-034955 | Ho-Chul Shin, Byung-Ha Oh |

The funders had no role in study design, data collection and interpretation, or the decision to submit the work for publication.

### Author contributions

LW, FB, Conception and design, Acquisition of data, Analysis and interpretation of data, Drafting or revising the article; AM, Acquisition of data, Drafting or revising the article; H-CS, CPT, B-HO, Drafting or revising the article, Contributed unpublished essential data or reagents; SG, Conception and design, Analysis and interpretation of data, Drafting or revising the article

## Additional files

### Supplementary file

• Supplementary file 1. Genotypes of *Bacillus subtilis* strains. All strains are derivatives of either *Bacillus subtilis* 1A700 (Bacillus Genetic Stock Centre) or *Bacillus subtilis* 168 ED.

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
