## [Decision Letter]

eLife posts the editorial decision letter and sponse on a selection of the published articles (subject to the approval of the authors). An edited version of the letter sent to the authors after peer review is shown, indicating the substantive concerns or comments; minor concerns are not usually shown. Reviewers have the opportunity to discuss the decision before the letter is sent (see review process). Similarly, the author response typically shows only responses to the major concerns raised by the reviewers.

Thank you for sending your work entitled “SMC condensin entraps chromosomal DNA by an ATP hydrolysis dependent loading mechanism in *Bacillus subtilis*” for consideration at *eLife*. Your article has been favorably evaluated by Richard Losick (Senior editor), a Reviewing editor, and three reviewers.

The Reviewing editor and the reviewers discussed their comments before we reached this decision, and the Reviewing editor has assembled the following comments to help you prepare a revised submission.

This paper from Gruber and colleagues presents outstanding experimental work that adds significantly to our mechanistic understanding of SMC biology. The work validates and characterizes a potentially broadly applicable assay to demonstrate topological association of protein complexes with DNA. The replisome-associated sliding clamp is used as a proof of principle and then the authors go on to show topological loading of the *B. subtilis* SMC complex onto DNA, and to demonstrate the requirements for ATP binding and hydrolysis on loading; the requirements for the accessory proteins ScpA and B and the requirement for ParB. Then the authors go on to develop an in vitro loading assay and then finally they show association of SMC complex with the ori region of the chromosome. Altogether, this is an important, impressive and extensive piece of work.

Specific comments requiring revisions:

1) The paper would benefit, especially for the broad *eLife* audience, from some speculation (even brief and tentative) about how topological entrapment might result in lengthwise condensation. Do the authors imagine that entrapment at ParB/parS transitions into loops? And if so, how? Similarly, does entrapment at *parB*/*parS* lead to diffusion via bead on a string followed by a transition into loops? Some discussion about possible models would strengthen the paper.

2) The Cys mutations in SMC and ScpA are not fully functional because they are synthetic lethal with *∆parB* in rich medium. This complicates the *∆parB* data a bit. It might be worth mentioning this.

3) Figure 5—figure supplement 1 legend could be re-written for clarity. “During step 2” is vague here.

4) Figure 6—figure supplement 1, the legend says anti-ParB antiserum. This should be anti-SMC.

I believe this figure is trying to say: “All proteins used in Figure 6 behave similarly in a standard ChIP experiment.” If yes, it would be helpful to state that explicitly.

5) In the Introduction, Figure 2 is referred to before Figure 1. Figure numbers should fit citation's order.

6) Figure 1: I found the cartoon that describes the protocol a little strange to begin with. It took me some time to realize the 'boxes' were the 'plugs' (a label would help), and the right hand side of the cartoon 'fades out' without really describing how the characterization is done-another appropriate panel or two would help the reader.

7) Figure 2 (and elsewhere): One needs a little faith and better labeling to be confident that the protein circle band is what they say it is (and it is relatively poorly resolved from the neighboring bands). I am convinced but the average reader would be helped with colored arrow labels and more explicit descriptions of what is what adjacent to the gel image.

8) It would have been nice to see a control showing that the nuclease benzonase is not contaminated with protease, otherwise the ‘+’ nuclease track data could have been explained by proteolytic digestion of the proteins? I guess this is pure recombinant nuclease that should be protease free-but it would be desirable to be assured that this is the case.

9) Figure 2—figure supplement 2: In panel A, why are bands 'fgh' not clear and not labeled?

10)Figure 3—figure supplement 1: The value of the Coomassie stained gel was not obvious, particularly because of the lack of label.

11) Figure 5 and accompanying text: I am intrigued to know why the authors didn't use cross-linkable proteins in this in vitro assay and a ScpA-TEV derivative to show that TEV cleavage releases SMC complex

12) Figure 6 legend: Presumably Smc is fused to the biotin peptide rather than the ligase BirA!

13) In the subsection headed “Smc–ScpAB rings physically associate with chromosomal DNA”, please explain briefly what the Avitag is.

14) Figure 6: It would help to have a map of the chromosome with positions of the genomic positions tested.

---

## [Author Response]

*1) The paper would benefit, especially for the broad* eLife *audience, from some speculation (even brief and tentative) about how topological entrapment might result in lengthwise condensation. Do the authors imagine that entrapment at ParB/parS transitions into loops? And if so, how? Similarly, does entrapment at* parB/parS *lead to diffusion via bead on a string followed by a transition into loops? Some discussion about possible models would strengthen the paper*.

To address these important points we have now included an additional figure (Figure 6) in the manuscript, which depicts two models for organization/lengthwise-condensation of chromosomes by Smc-ScpAB and *parB/parS*. In addition, we have added an extra paragraph in the Discussion called “How might entrapment of DNA at ParB/*parS* nucleoprotein complexes promote sister DNA segregation?” to elaborate and speculate on these models.

*2) The Cys mutations in SMC and ScpA are not fully functional because they are synthetic lethal with* ∆parB *in rich medium. This complicates the* ∆parB *data a bit. It might be worth mentioning this*.

We have added a sentence to the revised version of the main text (subsection entitled “ScpB and ParB proteins are essential for normal loading of condensin with chromosomes”) mentioning the possibility that the observed loading defect in *parB* mutants might be specific for the cys modified *smc* alleles and therefore might not be observable with wild-type Smc-ScpAB:

“The cysteine bearing *smc* allele (but not wild-type *smc*) causes growth defects when combined with *ΔparB* (Figure 4—figure supplement 1). Therefore, we cannot exclude the unlikely possibility that the decreased loading of Smc observed in *ΔparB* are due to the cysteine modifications in Smc and that chromosomal loading of wild-type Smc is not or much less affected by *parB* deletion.”

*3)*
Figure 5—figure supplement 1
*legend could be re-written for clarity.* “*During step 2*” *is vague here*.

This experiment was removed from the manuscript.

*4)*
Figure 6—figure supplement 1*, the legend says anti-ParB antiserum. This should be anti-SMC*.

*I believe this figure is trying to say:* “*All proteins used in*
Figure 6
*behave similarly in a standard ChIP experiment.*” *If yes, it would be helpful to state that explicitly*.

The legend now reads “anti-Smc antiserum”. We now declare the aim of the experiment presented in this figure supplement at the beginning of its figure legend (i.e. “All Smc variants used in Figure 5 behave similarly in standard ChIP experiments using anti-Smc antibodies.”)

*5) In the Introduction,*
Figure 2
*is referred to before*
Figure 1*. Figure numbers should fit citation's order*.

We have deleted the reference to Figure 2 from this early part of the main text.

*6)*
Figure 1*: I found the cartoon that describes the protocol a little strange to begin with. It took me some time to realize the 'boxes' were the 'plugs' (a label would help), and the right hand side of the cartoon 'fades out' without really describing how the characterization is done-another appropriate panel or two would help the reader*.

We thank the referees for pointing out the weaknesses of the illustration. We agree that the cartoon should start with cells in culture and show how the assay continues after digestion of chromosomal DNA until samples are analyzed with in-gel fluorescence. We have created a modified version of the cartoon describing all steps of the assay, improved the labelling and added more detailed description. We hope that the cartoon is now more clear and helpful to the reader.

*7)*
Figure 2
*(and elsewhere): One needs a little faith and better labeling to be confident that the protein circle band is what they say it is (and it is relatively poorly resolved from the neighboring bands). I am convinced but the average reader would be helped with colored arrow labels and more explicit descriptions of what is what adjacent to the gel image*.

Unfortunately, the linear and circular species are indeed closely migrating even after extensive optimization of the PAGE gels and run conditions. We have taken your suggestion and have now included colour-coded arrowheads to label all species on the gel images (Figure 2 and its supplements). The pictorial description of the species remains in the figure supplement. If needed, those can be moved or copied into the main figure.

*8) It would have been nice to see a control showing that the nuclease benzonase is not contaminated with protease, otherwise the ‘+’ nuclease track data could have been explained by proteolytic digestion of the proteins? I guess this is pure recombinant nuclease that should be protease free-but it would be desirable to be assured that this is the case*.

This is an important point. We thank the reviewer for pointing out this potential issue. The benzonase preparation is indeed purchased from a commercial provider (Sigma Aldrich) and should thus be largely free of proteases. To document that this is really the case we have now included two pieces of data in the revised version of the manuscript.

We have earlier noticed that flagellin is co-isolated during the chromosome entrapment assay, presumable due to its resistance to denaturation by SDS. Importantly, flagellin isolation is not affected by the addition of benzonase during cell lysis, as expected for an extracellular protein. This demonstrates that flagellin is not degraded by potential protease contaminations in the benzonase solution. We have included an extra sub-panel in Figure 1 to show normal flagellin isolation in a sample where entrapment of dnaN is lost due to benzonase treatment.

Independently, we have tested the effect of prolonged incubation of our protein extracts in the presence of benzonase. Coomassie staining and in-gel fluorescence analysis showed that the levels and migration pattern of Smc-HT species or proteins in general are not affected by excessive benzonase treatment (90 min). Please note that we have not done this experiment in the complete absence of benzonase as such protein extracts are only poorly resolved on our SDS-PAGE gels.

*9)*
Figure 2—figure supplement 2*: In panel A, why are bands 'fgh' not clear and not labeled?*

We have now labelled all visible bands in the gel image. Please note that species g, h, and i are not resolved from one another when a single cysteine residue is used for cross-linking of the Smc hinge. This is mentioned in the respective figure legend.

*10)*
Figure 3—figure supplement 1*: The value of the Coomassie stained gel was not obvious, particularly because of the lack of label*.

We used Coomassie staining to confirm uniform extraction of proteins during cell lysis in different samples. This is now made clear in the figure legend.

*11)*
Figure 5
*and accompanying text: I am intrigued to know why the authors didn't use cross-linkable proteins in this in vitro assay and a ScpA-TEV derivative to show that TEV cleavage releases SMC complex*.

We found that the expression of recombinant Cys-modified Smc protein in *E. coli* results in poor yields. Also, the cysteine cross-linking in vitro appears to be less specific than it is in vivo. We are optimistic to be able to solve these problems in the future. If successful, there will be several questions to be addressed with this assay, thus warranting an independent manuscript.

*12)*
Figure 6
*legend: Presumably Smc is fused to the biotin peptide rather than the ligase BirA!*

Completely right, thank you for pointing out this mistake. We have corrected the sentence.

13) In the subsection headed “Smc–ScpAB rings physically associate with chromosomal DNA”, please explain briefly what the Avitag is.

We now explain what the Avitag is in the main text in the subsection headed “Smc–ScpAB rings physically associate with chromosomal DNA fragments”:

“[…] a short Avitag peptide […] which gets biotinylated in *B. subtilis* when the biotin ligase *birA* is co-expressed (‘Smc-Avitag’).”

*14)*
Figure 6*: It would help to have a map of the chromosome with positions of the genomic positions tested*.

We agree that such a chromosomal map would be helpful to understand the figure more easily. We have added a simple scheme to Figure 5 and Figure 5—figure supplement 1.